# Wound Healing and Therapy in Soft Tissue Defects of the Hand and Foot from a Surgical Point of View

**DOI:** 10.3390/medsci9040071

**Published:** 2021-11-13

**Authors:** Wolfram Demmer, Heiko Sorg, Andreas Steiert, Jörg Hauser, Daniel Johannes Tilkorn

**Affiliations:** 1Klinik für Plastische, Rekonstruktive und Ästhetische Chirurgie, Handchirurgie, Alfried Krupp Krankenhaus Essen, 45276 Essen, Germany; Joerg.Hauser@Krupp-Krankenhaus.de; 2Abteilung für Handchirurgie, Plastische Chirurgie und Ästhetische Chirurgie, LMU Klinikum, 80336 München, Germany; 3Klinik für Plastische, Rekonstruktive und Ästhetische Chirurgie, Klinikum Westfalen, 44309 Dortmund, Germany; Heiko.Sorg@klinikum-westfalen.de; 4Fakultät für Gesundheit, Universität Witten/Herdecke, 58455 Witten, Germany; 5MeoClinic, 10117 Berlin, Germany; info@meoclinic.de

**Keywords:** wound healing, soft tissue defects, hand surgery, surgery of the foot, skin transplant, free-flap, microsurgery

## Abstract

Wounds and tissue defects of the hand and foot often lead to severe functional impairment of the affected extremity. Next to general principles of wound healing, special functional and anatomic considerations must be taken into account in the treatment of wounds in these anatomical regions to achieve a satisfactory reconstructive result. In this article, we outline the concept of wound healing and focus on the special aspects to be considered in wounds of the hand and foot. An overview of different treatment and dressing techniques is given with special emphasis on the reconstruction of damaged structures by plastic surgical means.

## 1. Introduction

Our hands and feet take up the key role in interaction of our bodies with the environment. The hand allows us to execute complex tasks ranging from fine motor skills to powerful grasps. At the same time, it provides a sense of touch, temperature, and proprioception [1]. The foot, with its highly sophisticated statics, allows us to walk upright and has a major impact on the entire musculoskeletal system. Special anatomical features of the soft tissues enable both hand and foot to fulfill these tasks and to resist various stress factors [2,3,4].

Superficial wounds and tissue defects on the hand and foot are frequent. They may be lengthy and tedious in healing, often resulting in long-lasting morbidity or disability. The reasons for wounds are various and range from physical, chemical, or thermal trauma to infections and ulcerations based on chronic diseases particularly connected to diabetes mellitus. They can be classified as acute or chronic considering the duration of healing [5].

Due to its exposed position and versatile use, the hand is at higher risk of sustaining trauma. Therefore, wound and tissue defects of the hand often originate from sharp or blunt trauma, animal bites, or burns [6,7]. If treatment is delayed, secondary infection often aggravates the clinical course. In the lower extremity, tissue defects more often result from underlying chronic diseases such as arterial malperfusion, diabetes, etc. as well as pressure-related ulcers. They can be aggravated by nutritional deficits of the patient [8]. In these instances, therapy must not only to be limited to conservative or surgical treatment of the wound itself but also focus on treatment of the underlying disease and risk factors [9].

Throughout life, all of the body’s tissues constantly lose cells and rebuild them. This turnover is driven by tissue stem cells that maintain tissue homeostasis [10]. Stem cells are unique in being able to self-renew and to generate more than one cell type [11].

Regarding the skin, tissue regeneration describes the constant specific substitution of tissue, as seen in the superficial epidermis or mucosa. Skin repair, on the other hand, represents an unspecific form of wound healing by fibrosis and scar formation. It represents the most common form of adult wound healing of the skin [12]. The process of skin repair always follows the same pattern of three phases, which are controlled by receptor-mediated signal transduction.

Hemostasis and inflammatory phase stand at the beginning of the healing cascade. The wound healing begins with the injury and lasts 1–3 days. Platelets invade the wound and cause homeostasis by forming a thrombus [13,14]. Macrophages and neutrophiles invade the wound for phagocytosis and secretion of growth factors and cytokines, thus creating an inflammatory environment [15,16,17,18]. Between day 2–21, the proliferation phase takes place. The wound surface begins to re-epithelialize from local keratinocytes at the wound edges as well as from epithelial stem cells in hair follicles or sweat glands [19,20,21]. The vascular system is restored by capillary sprouting [22,23]. Fibroblasts migrate into the wound and start to produce collagen III and other substances as a provisional wound matrix of connective tissue [24]. In the late period of the proliferation phase, this temporary tissue is replaced by granulation tissue, which contains a high number of fibroblasts, granulocytes, macrophages, and capillaries and collagen bundles [25,26,27,28]. The remodeling of the extracellular matrix is the final phase of wound healing and lasts from day 21 up to one year after the initial injury. The regenerative processes decrease while the wound matures leaving the mature wound tissue avascular and acellular [29]. Collagen III is replaced by the stronger collagen I. Finally, myofibroblasts cause tissue contraction, which helps to decrease the size of the developing scar.

Skin lesions can heal in three ways depending on the characteristics of the wound. It is important to choose the appropriate treatment method to avoid hypoxia, reduce edema, avoid infection, and clean the wound of foreign bodies. All of these adverse factors can cause insufficient wound healing and contribute to the formation of a chronic wound [30].

## 2. Pathways of Wound Healing

Wound healing is a complex process depending on various factors. Next to the overall health status of the patient, the cause of the lesion, its depth, and the involved underlying anatomical structures are all important considerations that need to be accounted for in the individual treatment plan. The treatment goal must be not only to restore the integrity of the skin but also to restore the function of the extremity.

Depending on kind and cause of the wound, healing can occur in three different ways: by primary, secondary, and tertiary intention [31].

### 2.1. Healing by Primary Intention

Primary healing of defects with minimal tissue loss can be archived by close approximation of the wound borders by surgical sutures or tapes in. In these instances, wound healing results in a thin scar. Primary closure of the skin can be acquired up to a maximum of 19 h after the wound [32]. Within 24 h, neutrophiles appear in the margins of the wound. After 24–48 h, the continuity of the epidermis is reestablished, sealing the wound against environmental impacts. Over the following weeks, fibroblasts proliferate and the inflammatory aspect of the healing gradually diminishes until the scar is covered again by an intact epidermis [33].

### 2.2. Healing by Secondary Intention

In cases of extensive tissue loss, either due to trauma or surgical intervention, primary closure of the wound is often unsuccessful. In open wounds, the defect is gradually filled by granulation tissue growing in from the wound margins and surface. Epidermal cells eventually spread from the wound edges to cover the granulation tissue and form scar tissue. Wounds left to heal by secondary intention require regular and intensive wound care measures. The degree scar contraction as well as adhesion to underlying structures is much higher than in wounds treated by primary intention.

### 2.3. Healing by Tertiary Intention

Tertiary healing is understood to be a delayed primary wound closure after 4–6 days or more, thereby interrupting the secondary healing of the wound. Reasons for a delayed closure can be infection of the wound, massive swelling of the surrounding tissue, necrosis that needs to demarcate before debridement, etc. Plastic reconstructive techniques may be necessary to archive an adequate wound closure.

Acute wounds usually heal in the sequence of inflammation, proliferation, and granulation as described above. If the wound fails to progress through these phases a chronic wound develops. This is usually due to factors that interfere with healing such as necrotic tissue, infection, or vascular malperfusion, which can cause edema and hypoxia.

If the wound is caused by trauma, the function of the involved extremity can be severely compromised due to the closed proximity of various underlying anatomical structures at the level of hand and foot. Concomitant injuries of bone, tendon, blood vessels, and nerves must always be ruled out before primary wound closure.

Wounds with substantial soft tissue defects may not be suitable for primary closure. When treating those wounds by secondary or tertiary intention as well as in plastic reconstructive approaches, the importance and particular nature of soft tissue cover of hand and foot must be taken into account in conservative as well as in surgical treatment.

## 3. Anatomical Characteristics of Soft Tissue on the Hand and Foot

The skin and soft tissue of the hand and foot is anatomically well adopted to meet the necessary elasticity and mechanical strength.

The palmar tissue of the hand needs to be above all pressure resistant while the dorsal skin must shift well over the underlying structures to allow the full range of motion.

The rigid skin on the palmar side of the hand lacks hair follicles and sebaceous glands but holds numerous sweat glands. It is characterized by papillary ridges at the fingertips and flexion folds where the skin is fixated to the palmar aponeurosis or tendon sheaths. The subcutaneous fat is arranged in compartments by perpendicular connective tissue to spread punctual pressure evenly over a large surface area decreasing the mechanical wear of the skin [34]. Numerous superficial lymphatic vessels run through the subcutaneous tissue of the fingers toward the palm, where they drain into collecting channels and continue towards the dorsum of the hand [35]. The dorsal skin of the hand is covered by nonrigid skin. Its thin subcutis has less fatty tissue and is only loosely connected to the extensor tendon sheaths to provide mobility of the skin envelope during joint extension or flexion.

The sole of the foot features unique characteristics to meet the mechanical requirements for weight bearing when standing and walking. The stratum corneum of the glabrous plantar skin is thicker than in other parts of the body; in addition, subcutaneous fat pads cushion the weight of the body. The fat tissue is fixed to the plantar fascia by strong septa of connective tissue to ensure optimal weight distribution to reduce plantar pressure and shear force. The pronounced sensibility of the sole of the foot protects against local trauma due to overstrain. A lack of sensibility on the contrary, for example, due to polyneuropathy, often leads to chronic ulcerations. In the dorsum of the foot, the mechanical properties are second to flexibility and adequate gliding of the skin over the extensor tendons. Accordingly, the skin is much thinner, flexible, and shifts well over the underlying tissue.

## 4. Wound Treatment of Hand and Foot Defects

Due to the importance of hand and foot in daily activities, the overall goal in wound management is to archive functional recovery. A thorough assessment not only of the wound but also of the patient as a whole should take place. Important issues are the period for which the wound already persists and the changes that it has undergone over time as well as known comorbidities or concomitant injuries of the patient. The treatment of chronic wounds often demands further diagnostics such as blood tests, vascular examination, and X-ray, CT, or MRI scans.

Since wounds of the hands often arise from injuries rather than underlying comorbidities, they tend to be acute rather than chronic. Therefore, suturing and healing by primary intention or plastic reconstruction of the tissue cover in wounds with exposed underlying structures such as tendons or bones are the common treatment approaches in these cases. In scrape wounds and superficial burns, on the other hand, conservative means, e.g., dressings, are applied and healing by secondary intention is sought.

One of the major causes of wounds and skin lesions of the foot is the diabetic foot. Its worldwide prevalence is 6.3% with the number of diabetes patients rising [36,37]. Diabetes mellitus is the underlying disease, but it is accompanied by neuropathy and vascular disorder [38]. Severity of comorbidities relate strongly with the outcome of the treatment of diabetic foot ulcer and must therefore be addressed and optimized [39,40,41]. Usually, clean wounds can be managed conservatively by dressings or negative pressure wound therapy (NPWT). An infection makes an antibiotic treatment necessary [42]. In case of an osteomyelitis or gangrene, a surgical intervention is often necessary with thorough debridement of the wound or even (partial) amputation of the foot [43].

### 4.1. Conservative Wound Management

For wounds ins general, but especially if healing by primary intention through surgical wound closure cannot be archived, the management of wound dressing is crucial. This applies to large acute skin defects as well as to chronic wounds.

The wound healing and re-epithelization are promoted by moist wound environment [44] but can also be impaired by excessive moisture due to massive wound exudate [45]. Therefore, important features of a proper dressing are the ability to absorb wound exudate and to prevent wound dehydration by maintaining a moist environment while allowing gas exchange. Furthermore, it needs to have adequate mechanical strength, be made of nontoxic and biocompatible material, nonadherent, comfortable to wear, and easy to be removed. Beyond that, antibacterial and antifungal qualities may be desirable. Ideally, the wound dressing contributes to rapid wound healing at a reasonable cost [46].

Wound dressings can roughly be classified by their function, e.g., absorbent, (non)adherent, debriding, antibacterial, occlusive [47], the material they consist of, e.g., collagen, hydrocolloid, alginate [48], and their physical form, e.g., tissue, foam, film, ointment, or gel [49].

Primary dressings have direct contact to the wound; secondary dressings cover the primary dressing. Dressings with a central absorbent and an outer adhesive part are called island dressing. According to their interaction with the wound, dressings can be categorized as passive, interactive, and bioactive [50].

Traditional dressings such as cotton, natural or synthetic gauze, and bandages are dry dressings and can be used as primary or secondary dressing. They qualify as passive or nonocclusive dressing. The transport capability of these materials for gas and moisture are high. In chronic wounds with high exudation levels, they can adhere to the wound and cause severe pain and damage during removal. Passive dressings are therefor rather suitable for dry wounds [51].

In the last several decades, modern active wound dressings were developed that mainly preserve or create a moist wound environment, thereby facilitating wound healing and decreasing the risk of bacterial infection [52].

Some of the most frequently used modern wound dressings are hydrocolloid dressings. They adhere to moist and dry sites and are used mainly for pressure sores, ulcers, uncomplicated small burns, and traumatic injuries.

Alginate dressings consist of polysaccharides obtained from alginic acid produced by seaweeds [53]. Alginates are highly absorbent and are therefore well suited for exuding wounds. Upon contact with the exudate of the wound, they form a protective film of gel that creates an optimal environment for wound healing in terms of moisture and temperature [54].

Hydrogels are hydrophilic dressings made of synthetic polymers. They contain 70–90% of water, and hydrogels cannot absorb much fluid and are mainly used for light or moderately exuding wounds. In dry or necrotic wounds, they rehydrate the tissue and promote autolytic debridement.

Semipermeable adhesive film dressings can be used as primary dressing, e.g., in (sub)total amputations of the fingertip [55], or as secondary dressing for patients with high sensitivity to adhesive plaster. Being semipermeable, the film allows water vapor and gases to pass through, but wound exudate cannot. Accumulation of the latter under the dressing can occur leading to maceration and unpleasant odor [56].

Foam dressings consist of porous polyurethane foam. They are highly adsorbent and have a high moisture vapor transmission rate [57]. Foam dressings insulate the wound and keep it moist and are comfortable to wear for the patient.

A variant of occlusive dressing is the vacuum-assisted negative pressure wound therapy (NPWT). The rate of granulation and epithelialization is promoted by applying subatmospheric pressure to the wound. NPWT can be used to promote secondary closure of a wound or as a bridge to surgical wound closure by skin graft or flap [58,59]. Next to clinical use, small single-use devices have been introduced to ambulant wound care.

Bioactive dressings not only create an optimal environment for wound healing but also deliver bioactive substances to the site of healing. Biologically active dressings include tissue engineered products usually combining for example collagen, chronically acid, chitosan, alginates, or elastin [60,61,62,63]. Using natural or synthetic components, multiple types of smart wound dressings have been developed to mimic the native environment of skin tissue [64,65].

Current development and research in smart dressings try to integrate self-adjust treatment into the complex process of wound healing. In so-called smart dressings, bioactive components are sensitive to changes in the wound environment such as ROS (reactive oxygen species), cytokines, or enzymes and are activated accordingly by releasing growth factors, anti-inflammatory agents, or different small molecules [66,67,68].

### 4.2. Surgical Wound Management of the Hand

Clean wounds without greater tissue defect can be closed with nonabsorbable monofilament sutures. On the palmar skin, single stitches should be used; at the dorsal aspect, wounds can also be sutured continuously. In small children, absorbable sutures should be used.

The palmar skin of the hand has a good regenerative capacity. Minor superficial defects or surgical wounds without involvement of underlying structures will heal via secondary intention within weeks. Larger defects on the hand usually need some kind of reconstructive procedure to ensure adequate healing, protect exposed tendons neurovascular structures or bones, and prevent scar contractions.

In the past, defects at the tip of the fingers such as (sub)total amputation of the distal phalanx were treated by various local flaps such as the Tranquilli-Leali-Flap [69]. Currently, those injuries are usually treated by semiocclusive dressing of the wound. After initial inspection and debridement of the wound in local anesthesia, a semiocclusive dressing of self-adhesive polyurethane is applied. The dressing should stay on until re-epithelization is complete after approximately 3–7 weeks. Using this conservative treatment, an extensive regeneration of a sensible fingertip can be archived.

Large defects in the palm and proximal fingers should be treated by plastic reconstructive means in order to prevent joint contractions. If the wound is superficial and the tendon sheath intact, a skin graft is the simplest way to close the wound. To ensure a durable skin coverage, a full thickness skin graft of similar skin should be used, especially in the areas of grip. Possible donor sites are the ulnar edge of the hand, where a spindle of approx. 5 × 1.5 cm can be harvested with minimal donor site morbidity. Another donor site is the in-step area of the foot. This way, the harvested piece of skin can be larger, but the donor site defect needs to be closed with a split skin graft. It is important not to impair the weight-bearing sole of the foot.

Exposed functional structures such as tendons or bone make a reconstruction including subcutaneous tissue necessary. For these cases, a wide variety of local, distant, or free flaps have been designed. For smaller defects of the back of the hand, local rotation or transposition flaps are usually sufficient. For a proximally pedicled flap, the entire skin of the dorsum can be raised and shifted.

In lesions of the palmar thumb, not only do the defect needs to be closed, but sensitivity must also be restored. The Foucher flap is an island flap of the dorsal proximal phalanx of the index, pedicled at the first dorsal metacarpal artery (Figure 1A–C). If the terminal branches of the superficial radial nerve are included, this flap provides an excellent sensate cover for the palmar thumb [70].

The lesion at the finger level with exposed functional structures can, for instance, be covered via a cross-finger-flap. For palmar defects, the pedicled flap is raised from the dorsal aspect of the neighboring finger; for dorsal defects, a reverse cross finger flap can be raised also from the dorsum of the neighboring finger. The donor site is closed by split skin graft. To prevent adhesion of the extensor tendons and the skin graft, meticulous preparation of the tissue layers is important [71]. For medium sized defects, especially on the dorsum of the finger or the proximal palmar aspect of the finger, a flap distally based on the dorsal metacarpal artery (DMCA) is a viable option [72].

Larger defects of the hand need a plastic reconstruction either via pedicled distant flaps or free flaps. The pedicled radial forearm flap poses a locoregional option for dorsal and palmar lesions of the hand [73]. The perfusion of this flap is retrograde, so an intact ulnar artery and palmar arterial arch are required. The Allen test should be performed preoperative to ensure the blood supply of hand and flap.

Based on the same concept, a retrograde perfused flap nourished by the dorsal interosseous artery provides an adequate soft tissue replacement for substantial defects of the dorsum of the hand [74] (Figure 2A–D).

For special indications, the pedicled groin flap remains a valuable option, especially in patients with severe arteriosclerosis or extensive forearm trauma. A disadvantage of this flap is the immobilization of the hand and arm until the flap can be safely detached from the groin, approximately three weeks after the primary operation [75].

If local options for reconstruction are not viable or the donor side morbidity would be disproportionate, free microvascular flaps can be used. Usually, fascial, adipocutaneous, or fasciocutaneous flaps offer a sufficient tissue coverage. An example out of the multitude of possible free flaps is the anteriolateral tight flap (ALT). The ALT is a relatively thin fasciocutaneous perforator flap that can be harvested with minimal donor site morbidity [76]. In obese patients, the subcutaneous fat can be thicker than desired. Especially when used for reconstruction of the palmar hand, voluminous flaps can cause problems in fist closure. Primary or secondary thinning of the flap may be necessary.

Where fine coverage of exposed structures is desired, e.g., the finger or palmar areas of grip, the arterialized venous flap poses an advantageous alternative (Figure 3A,B). The flap is preferably taken from the forearm together with a subcutaneous vein. Both ends of this vein are then anastomosed to artery and vein at the recipient site, respectively [77].

### 4.3. Surgical Wound Management of the Foot

The general considerations in wound treatment of the foot resemble those described for the hand; clean fresh wounds can be treated accordingly.

Yet, in contrast to the wounds of the hand, most wounds of the foot are more commonly chronic ulcerations rather than caused by acute trauma or tumor excision. Common diseases that cause ulcers at the level of the lower leg and foot are diabetes mellitus, peripheral artery disease, or chronic venous insufficiency often aggravated by comorbidities such as immunosuppression due to steroid use, renal impairment, autoimmune diseases, dermatological diseases, or paralysis, etc. An impaired general or local condition can greatly complicate and prolong the healing process. Predestined locations for ulcers of the foot are areas of highest pressure such as the heel and the ball of the big and small toe.

In general, chronic wounds show a certain degree of necrotic or infected tissue and are usually contaminated by microbes. The first step is thorough (surgical) debridement and appropriate wound care. Due to the unique mechanical qualities of the sole of the foot, debridement should spare as much skin as possible, saving even small pieces of original sole tissue.

Negative pressure wound therapy (NPWT) can be helpful in preparing chronic wounds for reconstructive coverage. It is a widely accepted and used treatment of diabetic foot and pressure ulcers. Its ability to reduce wound size and promote granulation tissue growth has been shown [59]. By continuous suction, it drains wound secretion and stimulates local blood circulation, thereby inducing granulation tissue by inducing mild hypoxia angiogenesis [78]. With this method, even deep cavities can fill up with granulation tissue, preconditioning the wound base for possible skin transplantation or leaving it for closure by secondary intention.

Especially at the dorsal foot, splitskin graft is a fast and reliable method to restore skin continuity as long as no functional structures are exposed. On the plantar side of the foot, skin grafting is feasible in areas without mechanical strain, e.g., the instep area of the foot. Wounds in the weight-bearing zones such as the heel or the ball of the big and small toe must be treated by pedicled of free tissue transfer. In selected cases, the wounds should be left to conservative healing due to the patient’s general condition, e.g., bedridden patients.

For reconstruction of the sole of the foot, techniques that ensure sensibility and withstand the mechanical strain should be used. The shifting of the transplanted tissue must be minimal to prevent unstable scars, ulceration, or unsteadiness when walking. Local random pattern flaps such as the VY, rotation, or rhomboid flaps can be used successfully for smaller defects in the ball area of the foot. Safety as well as potential flap seize can be augmented when using delayed techniques. Defects from medium to large seize in the weight-bearing areas of the foot, such as the heel or forefoot, can be treated by pedicled or free medial plantar (instep) flap (Figure 4A–C). The innervated flap is pedicled to the medial plantar artery. It can be 8–10 × 6 cm in diameter with the maximum of the flap outline confined by the weight-bearing areas. Its rotational arc allows coverage of the heel, midplantar area, and lateral edge of the foot. The donor area can be closed via skin graft [79].

In selected cases, e.g., in patients with impaired circulatory status, defects of the forefoot can be closed by filet flaps of the toes. Since the sacrifice of one or more otherwise unharmed toes is necessary this “spare part” procedure should be reserved as a last resort if, for example, a more proximal amputation would otherwise be necessary [80].

For extensive defects or if local options for reconstruction are not available, a free microvascular tissue transfer becomes necessary. The main problem with any flap from another area of the body is the higher shifting of its subcutaneous tissue compared to the sole of the foot. This can lead to shearing of the tissue layers, which can result in the formation of pseudobursa or detachment of the flap from its base. To reduce shearing and the formation of pseudobursa, pure muscle flaps such as the latissimus dorsi flap covered by split skin graft are preferred. The contact of skin graft to muscle is fixed and does not glide thereby minimizing the shifting effect. However, split skin graft is not resistant to pressure and friction, leaving it vulnerable resulting in frequent superficial wounds especially in zones of pressure. Mere muscle flaps lack sensation, even though the restoration of some sensation by coaptation of a native sensory nerve to the motor nerve of a skin grafted muscle flap has been reported [81].

A more common approach to restore sensation to the sole of the foot is the use of fasciocutaneous perforator flaps such as the ALT with coaptation of its sensible nerve to the lateral femoral cutaneous nerve or branches of the sural nerve. The flap can be thinned to 3–6 mm intraoperatively to reduce later shearing of the tissue [75], or it can be thinned in a separate operation via liposuction after completion of the healing process. Up to a width 10–12 cm of the ALT the donor site can be closed primarily.

The after care in patients that underwent plastic-reconstructive surgery of the foot is important not only to avoid postoperative problems in wound healing but also to prevent long-term complications due to over or improper use of the foot. Therefore, physiotherapeutic walking training and an early orthopedic shoe care to relieve pressure and minimize friction on the reconstructed site is paramount.

## Figures and Tables

**Figure 1 medsci-09-00071-f001:**
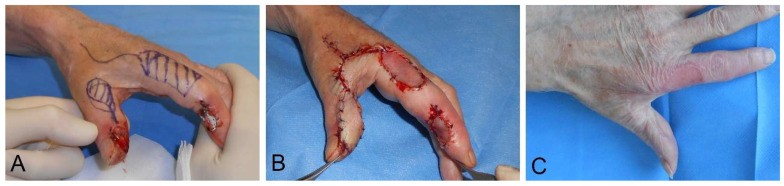
Reconstruction of a palmar thumb defect following a third degree burn by high voltage trauma with a Foucher flap. (**A**) Preoperative flap design. (**B**) Postoperative result. (**C**) Long-term result after >6 months.

**Figure 2 medsci-09-00071-f002:**
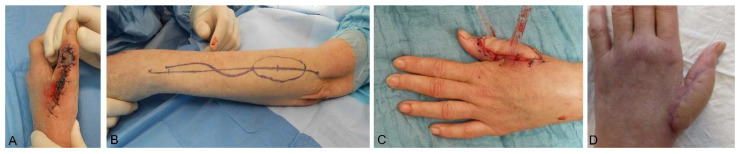
Reconstruction of a dorsal thumb defect following a purulent extensor tendon synovitis with subsequent soft tissue defect with a pedicled dorsal interosseous artery perforator flap. (**A**) Unstable primary closure after initial debridement of the extensor tendon. (**B**) Preoperative flap designed. (**C**) Postoperative resulting. (**D**) Long-term result after >6 months.

**Figure 3 medsci-09-00071-f003:**
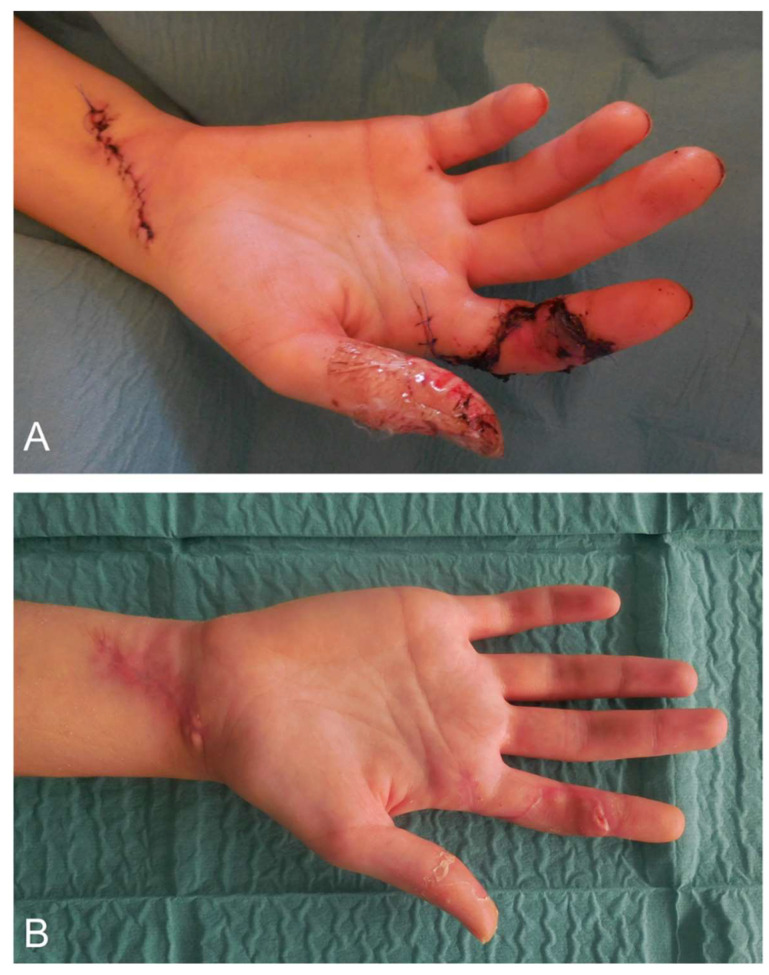
Arterialized venous flap on the radiopalmar side of the index finger for reconstruction of a radiopalmar soft tissue defect of the index finger. (**A**) Postoperative coverage by an arterialized venous flap and donor site at the palmar distal forearm. (**B**) Long-term result after >2 months.

**Figure 4 medsci-09-00071-f004:**
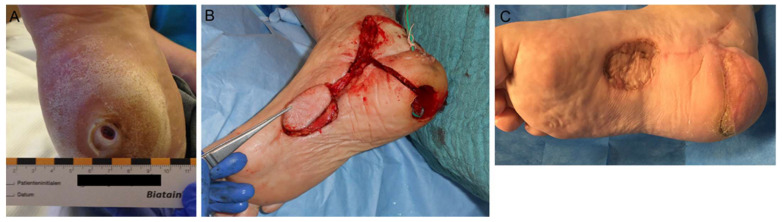
Chronic ulcer of the heel, coverage by pedicled medial plantar artery (instep) flap. (**A**) Preoperative finding at the right heel. (**B**) Intraoperative preparation of the instep flap. (**C**) Long-term result after >4 months.

## Data Availability

Data is contained within the article or references.

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
