# Peer review of "Wound Healing and Therapy in Soft Tissue Defects of the Hand and Foot from a Surgical Point of View"

_medsci, 2021, doi:10.3390/medsci9040071_

Round 1

Reviewer 1 Report

The manuscript can be now considered for publication.

Reviewer 2 Report

The article has been improved according to the requested indications.

This manuscript is a resubmission of an earlier submission. The following is a list of the peer review reports and author responses from that submission.

Round 1

Reviewer 1 Report

This manuscript by Demmer et al. reviewed the wound healing and therapy in soft tissue defects of the hand and foot and has discussed the anatomical characteristics of soft tissue on the hand and foot, basic principles in wound healing, wound healing and reconstruction of the soft tissue cover. This review article fits the scheme of the journal and can be considered for publication after addressing the following issues.

  1. The introduction part and the abstract are too short and there is no reference in the introduction.
  2. Some advanced wound dressings should be introduced.

Author Response

Dear reviewer,

thank you for your comments on the above mentioned article.

Concearning the issues mentioned:

1) Introduction part was extended, References were added.

2) The aim of the article was clarified. Focus lies on surgical therapies, therefore indulgement in conservative wound dressings would blow the scope of this article.

Thank you for revising.

Sincirely

Dr. Demmer

Reviewer 2 Report

1)The authors should elaborate more  the concept of tissue repair in the introduction.

2)It would be interesting to evaluate the various therapies according to the phases of tissue repair

3)The authors have predominantly described surgical techniques but the title refers to various tissue repair techniques. Currently, there are  different technologies that can be used.

The authors should add description of:

  • current dressings e new generation biomaterials
  • ultra-portable negative pressure therapies
  • stem cells
  • new generation grafts
  • photobiomodulation
  • and more.

4)Results should be structured according to the various therapies

4)References are few for a review and they should be improved

Please consider some of these articles for example:

Chouhan D, Dey N, Bhardwaj N, Mandal BB. Emerging and innovative approaches for wound healing and skin regeneration: Current status and advances. Biomaterials. 2019 Sep;216:119267. 

Leong S, Lo ZJ. Use of disposable negative pressure wound therapy on split-thickness skin graft recipient sites for peripheral arterial disease foot wounds: A case report. Int Wound J. 2020 Jun;17(3):716-721.

Farahani M, Shafiee A. Wound Healing: From Passive to Smart Dressings. Adv Healthc Mater. 2021 Jun 26:e2100477

Fuchs E, Blau HM. Tissue Stem Cells: Architects of Their Niches. Cell Stem Cell. 2020 Oct 1;27(4):532-556. 

Las Heras K, Igartua M, Santos-Vizcaino E, Hernandez RM. Chronic wounds: Current status, available strategies and emerging therapeutic solutions. J Control Release. 2020 Dec 10;328:532-550. 

Author Response

Dear reviewer,

thank you for your comments on the above mentioned article.

Concearning the issues mentioned:

1) Concept of tissue repair was added to introduction

2) The article aims to show possible treatment Options for wound healing disorders from a surgical point of view (this was clarified in the revised title of the article) considering especially the unique texture of hand and foot tissue . Evaluation according to phases of tissue repair would unfortunalty blow the Focus of this article.

3) see 2)

4) see 2) I included some of the papers proposed.

Thank you for revising.

Sincirely

Dr. Demmer